# Positive Youth Development and Adolescent Depression: A Longitudinal Study Based on Mainland Chinese High School Students

**DOI:** 10.3390/ijerph17124457

**Published:** 2020-06-21

**Authors:** Zheng Zhou, Daniel T.L. Shek, Xiaoqin Zhu, Diya Dou

**Affiliations:** 1Research Institute of Social Development, Southwestern University of Finance and Economics, Chengdu 611130, China; zhouzheng@swufe.edu.cn; 2Department of Applied Social Sciences, The Hong Kong Polytechnic University, Hong Kong, China; xiaoqin.zhu@polyu.edu.hk (X.Z.); diya.dou@polyu.edu.hk (D.D.)

**Keywords:** positive youth development, depression, mental health, adolescence, longitudinal study

## Abstract

There are several limitations of the scientific literature on the linkage between positive youth development (PYD) attributes and adolescent psychological morbidity. First, longitudinal studies in the field are limited. Second, few studies have used validated PYD measures to explore the related issues. Third, few studies have used large samples. Fourth, limited studies have been conducted in mainland China. In this study, we conducted a longitudinal study using two waves of data collected from 2648 junior high school students in mainland China. In each wave, participants responded to a validated PYD scale (Chinese Positive Youth Development Scale: CPYDS) and other measures of well-being, including the 20-item Centre for Epidemiologic Studies Depression Scale (CES-D). After controlling for the background demographic variables, different measures of CPYDS (cognitive–behavioral competence, prosocial attributes, general positive youth development qualities, positive identity, and overall PYD qualities) were negatively associated with CES-D scores in Wave 1 and Wave 2. Longitudinal analyses also revealed that PYD measures in Wave 1 negatively predicted Wave 2 depression scores and the changes over time. The present findings highlight the protective role of PYD attributes in protecting adolescents from depression.

## 1. Introduction

Adolescent developmental issues and mental health problems have drawn much attention from the public and helping professionals. Existing research has revealed some prominent adolescent mental health issues, including depression, suicide, and anxiety-related problems [1]. Lim et al.’s meta-analysis, involving 686,672 children and adolescents, revealed that the aggregate lifetime and 12-month prevalence rates of suicide attempts, suicidal plans, suicidal ideation, non-suicidal self-injury, and deliberate self-harm were 4.5%, 7.5%, 14.2%, 19.5%, and 14.2% in adolescents, respectively [2]. In addition, internet addiction, gaming problems, pathological gambling, and social media addiction have rapidly increased in recent decades [3,4,5]. UNICEF warned that one-fifth of adolescents in the world have mental health problems, which could hinder adolescent adjustment and create unfavorable consequences in adulthood [6]. The WHO also indicated that mental health problems have contributed to around 16% of the disease and injury in adolescents aged between 10 and 19 [7].

A salient adolescent mental health problem is depression. According to the WHO [7], depression is a major form of adolescent psychological morbidity. High prevalence rates of adolescent depression have been reported in epidemiological studies globally. The National Institute of Mental Health [8] in the United States showed that 13.3% of the students aged between 12 and 17 had major depressive disorder. Balazs et al. [9] reported prevalence rates of 29.2% and 10.5% for subthreshold depression and depression, respectively, in adolescents in 11 European countries. Similar findings also have been observed in Sweden [10] and Australia [11]. In Asian countries, a meta-analysis based on 42,347 Chinese children showed that the prevalence of depressive symptoms amongst primary school students was 17.2% [12].

Cross-culturally speaking, individuals in Asian countries were more likely to suffer from depressive symptoms than their counterparts in Western countries. For example, Stewart et al. [13] found that, while 38% of Hong Kong adolescents demonstrated mild or moderate levels of depressive symptoms, the figure was 25% for adolescents in the United States. Steptoe [14] also revealed that around 38% of students in Asian regions (e.g., Japan, Korea, and Taiwan) showed depressive symptoms, while less than 20% of students in northwestern Europe and the United States showed depressive symptoms. Similar findings have been found in other cross-cultural studies [15,16] and meta-analysis studies [2]. This difference may be attributable to psychosocial factors such as higher levels of parental expectations and academic pressure [17] but lower social support from parents and friends [16] in adolescents in Asian countries than in Western countries. Besides, as Asian children are usually socialized to show higher levels of self-control, obedience, and respect for authority, as well as avoiding creating burdens for other people, they are less likely to seek family and social support when encountering emotional distress [18]. Furthermore, recognizing one’s shortcomings, achieving self-improvement, and displaying modesty are more advocated in Asian traditions than in Western values. Therefore, Asian students may be more prone to have self-criticism and negative self-evaluation, which might lead to a higher risk of developing depression [14]. Furthermore, although some preliminary findings suggest some common genetic factors of depression in Asian and non-Asian people [19], Chen et al. found that ecological factors contributing to adolescent depression are common in Chinese and non-Chinese cultures [20].

Different approaches have been used to understand adolescent depression, including genetic, neuroscience, Freudian, cognitive, and socio-cultural perspectives [21,22,23]. We can highlight three observations from these perspectives. First, while psychological approaches commonly focus on intrapersonal antecedents of adolescent depression, socio-cultural models examine societal factors (such as poverty and health inequalities) and social constructions in adolescent depression. With the emergence of the ecological perspective, it is argued that there is a need to look at the impacts of different systems on adolescent emotional disorders [24,25]. Second, existing theories of depression have been primarily based on adults, assuming that adult models are equally applicable to adolescents. Third, most of the depression theories look at depression from a “deficit” perspective. Except for models based on humanistic, existential, and positive psychology, few theorists consider the role of psychosocial competences and developmental assets in adolescent depression.

The development of humanistic psychology, an ecological perspective, and positive psychology has spawned views arguing the importance of the developmental contexts and assets of adolescents to prevent mental issues. These views are commonly subsumed under the rubric of the “positive youth development” (PYD) approach, which highlights the importance of both internal and external developmental assets to healthy adolescent development [26]. In their review of the current perspectives of PYD models, Shek et al. [27] outlined and compared several theoretical models, including developmental assets proposed by Benson [28], Lerner’s six Cs model [29], social and emotional learning (SEL) [30], the “being” perspective [31], and the 15 PYD attributes summarized by Catalano et al. [32], in terms of the similarities and differences between these models.

Holding a specific PYD attribute perspective, Catalano et al. reviewed 75 PYD programs and identified 25 PYD programs that showed beneficial program outcomes [32]. These 25 effective PYD programs highlighted 15 PYD attributes (see Table A1 in Appendix A). Empirical research has found that these 15 PYD attributes are protective factors against adolescent depression [33,34]. For example, existing studies have underscored the contribution of social bonding to positive adolescent development [35,36,37]. Similarly, empirical studies also highlighted the important roles of resilience [38,39], psychosocial competences such as emotional and cognitive competence [40,41], and positive self-perceptions, such as self-efficacy and positive identity [22,42,43], in helping adolescents overcome life adversities and establish positive self-concepts. Therefore, promoting PYD attributes would benefit positive adjustments and prevent mental issues and problems in adolescents. Empirically, several review studies highlighted the effectiveness of the PYD perspective in promoting positive developmental outcomes and reducing adolescent problem behavior [44,45,46].

Although existing research has underscored the positive effects of PYD attributes on adolescent development, several research gaps exist in the literature on the linkage between PYD attributes and adolescent depression. First, most of the available studies have been conducted in Western contexts, particularly in the United States. As the impact of PYD is contingent on the developmental context in which culture is an important dimension [26], there is a need to conduct more research in non-Western contexts. Studies in non-Western communities can help to ascertain the generalizability of the utility of PYD programs [47]. Second, studies in the field are dominated by cross-sectional studies. While cross-sectional studies are useful, they cannot ascertain the cause and effect relationship between PYD and adolescent depression over time. Hence, more longitudinal studies in this area are needed. Third, as many studies focused on the relationship(s) between one or two aspect(s) of PYD and adolescent morbidity, the employment of multiple PYD measures, such as cognitive–behavioral competence, prosocial attributes, identity-related attributes, and global PYD features, would give a more complete picture. Fourth, as some of the existing studies were based on small samples, the employment of a larger sample would enhance the generalizability of the findings and ensure the power of statistical analyses. Fifth, there are few longitudinal studies examining the influence of PYD attributes on adolescent depression in the Chinese context. Although there are some studies on the influence of PYD attributes on the mental health of early adolescents [48], late adolescents, and emergent adults [1,49], related studies are almost non-existent in mainland China. Using the search term “positive youth development and adolescent depression” in a computer search using PsycINFO in June 2020 showed that there were 840 citations. When we added “Chinese” in the search, the number of citations became 26.

With reference to the above research gaps, the present study attempted to examine the relationship between PYD attributes and adolescent depression, utilizing a short-term longitudinal study, with data collected in two waves separated by one year (i.e., Wave 1 and Wave 2 data). We examined the following research questions in this study:

Research Question 1: Are there cross-sectional associations between PYD attributes and adolescent depression? Based on the literature, the general hypothesis was that PYD attributes would have negative cross-sectional relationships with adolescent depression in Wave 1 and Wave 2 (Hypothesis 1). Although cross-sectional analyses are problematic in identifying the cause and effect relationship, we still included the cross-sectional analyses because related research in the Chinese context is very sparse.

Research Question 2: Are there longitudinal associations between PYD attributes and adolescent depression? Based on the literature, the general hypothesis was that PYD attributes in Wave 1 would have negative longitudinal relationships with adolescent depression in Wave 2 (Hypothesis 2).

Research Question 3: Are PYD attributes related to change in adolescent depression over time? Based on the literature, the general hypothesis was that PYD attributes would be negatively related to change in adolescent depression over time (Hypothesis 3).

## 2. Materials and Methods

### 2.1. Participants and Procedures

To address the research questions, we designed a longitudinal study spanning over two years. In the context of the Tin Ka Ping P.A.T.H.S. Project [50], we invited four junior high schools in mainland China to join a short-term longitudinal study with two waves to study the psychosocial adjustment of Chinese high school students and the related determinants. In the 2016–2017 school year, Grade 7 (*N* = 1362) and Grade 8 (*N* = 1648) students participated in the study. In each wave, students completed a questionnaire assessing different PYD attributes (such as resilience, psychosocial competence, and spirituality), psychological well-being, risk and problem behavior, materialism, egocentrism, academic values, and academic anxiety. The study was approved by the Human Subjects Ethics Subcommittee at The Hong Kong Polytechnic University. Before data collection, school, parental, and student consent was obtained. When collecting data, researchers informed all student participants about the study aim and the confidentiality and anonymity principles of data collection and analysis.

In the 2017–2018 school year, 1305 and 1343 students of the original Grade 7 and Grade 8 students joined the study in the second year, respectively, with attrition rates of 4.19% and 18.51% for the original Grade 7 and Grade 8 students, respectively. In the matched sample (*N* = 2648), 1513 were male, 1109 were female, and 26 students did not report their gender. The mean age of the matched sample was 13.12 ± 0.81 years in Wave 1. Regarding the marital status of the parents, most students (*N* = 2225) were in intact families (i.e., their parents were in their first marriage). In total, 401 students reported that their families were non-intact, with parents separated, divorced, or re-married. In the present study, background socio-demographic variables included student age, gender, family intactness, whether the respondent lived with the father (Yes = 2202), and whether the respondent lived with the mother (Yes = 2368).

### 2.2. Measures

In this study, we used a questionnaire to assess the psychosocial development of adolescents in mainland China. The measures included validated tools on PYD, psychological well-being, problem behavior (e.g., internet addiction and suicidal behavior), materialism, egocentrism, academic achievement, and academic anxiety. The details of the measures were reported elsewhere [4,5]. In this study, we focused on the relationships between PYD attributes and adolescent depression.

#### 2.2.1. PYD Attributes

We used the 80-item Chinese Positive Youth Development Scale (CPYDS) to assess PYD attributes in adolescents in this study. It is an indigenous Chinese measure of PYD, which has been validated in Chinese adolescents [51,52]. Based on the 15 PYD attributes identified in the successful PYD programs [32], the CPYDS included 15 subscales, including measures of psychosocial competence, positive self-related attributes, spirituality, and prosocial involvement. Previous studies showed that there were 15 primary PYD factors and four higher order factors in CPYDS. The first higher order factor is “Cognitive-Behavioral Competence (CBC)” which comprises “Cognitive Competence”, “Behavioral Competence”, and “Self-determination” subscales. The second higher order factor is “Prosocial Attributes (PA)” which is made up of “Prosocial Norms” and “Prosocial Involvement” subscales. The third higher order factor is “General PYD Qualities (GPYD)” which includes “Bonding”, “Resilience”, “Social Competence”, “Recognition for Positive Behavior”, “Emotional Competence”, “Moral Competence”, “Self-Efficacy”, and “Spirituality” subscales. The fourth higher order factor is “Positive Identity (PIT)” which is composed of “Clear and Positive Identity” and “Beliefs in the Future” subscales. All items used a six-point reporting scale (“1 = strongly disagree”; “6 = strongly agree”). A composite score for each higher order factor and an average of total PYD score for all items (TPYD) were computed. In the present study, all subscales demonstrated acceptable internal reliability in both waves (see Table 1). Besides, the mean item–item correlation and item–total correlation values were acceptable. Although self-efficacy demonstrated relatively low reliability (0.54), it was included in the analyses for three reasons. First, some references suggest that this reliability level for scales with few items is marginally acceptable [53,54]. Second, the mean inter-item correlation (0.37 and 0.42 in Wave 1 and Wave 2, respectively) and item–total correlation (0.37 and 0.42 in Wave 1 and Wave 2, respectively) were acceptable. Third, the main analyses in the present study used higher order factors showing good reliability. Fourth, the 15 PYD attribute framework has a relatively strong conceptual foundation. The means that the PYD primary factors ranged between 4.43 and 5.48 in Wave 1, and between 4.55 and 5.42 in Wave 2. The “Clear and Positive Identity” and “Spirituality” subscales demonstrated the lowest and the highest mean scores, respectively, in both waves.

#### 2.2.2. Depression

We used the Center for Epidemiologic Studies Depression Scale (CES-D) [55] to measure depression. As a self-report measure of depressive symptoms, the CES-D has been widely used in different age groups in diverse cultural contexts, including Chinese communities [56]. Participants reported the frequency of depressive symptoms during the past week on a four-point Likert Scale (1 = “rarely or none of the time”; 4 = “most of or all of the time”). In the present study, the CES-D also showed good internal consistency with Cronbach’s α > 0.85 in both waves (see Table 1).

### 2.3. Analysis Plan

Shek [57] pointed out that there are different ways of analyzing longitudinal data, including correlational analyses (such as longitudinal and partial correlation analyses), multiple regression analyses (such as hierarchical multiple regression controlling for extraneous variables), and structural equation modeling (SEM) involving latent variables. Despite the advantages of SEM, we used hierarchical multiple regression analyses in this study to answer the research questions, with two justifications. First, this approach has been commonly employed by researchers to examine research questions based on longitudinal data [58,59]. Second, researchers have used this approach to examine the influence of PYD attributes on adolescent mental health [42].

Cohen et al. [58] highlighted that in analyzing longitudinal data, it is important to remove the effects of extraneous variables. In this study, extraneous variables included both background socio-demographic variables (such as gender and age) and the initial level of the criterion variable. For concurrent predictions, we used PYD attributes at a time point to predict depression scores at that time point after controlling for the background socio-demographic variables in the first block. To look at the prediction of Wave 1 predictors on Wave 2 depression scores, we first included Wave 1 extraneous socio-demographic variables in the first block. In the second block, one Wave 1 PYD measure, including CBC, PA, GPYD, PIT, or TPYD, was added to the analysis. If a significant F change and standardized regression coefficient were observed at this block, we would interpret that specific Wave 1 PYD predictor as having had an effect on Wave 2 depression scores.

Besides looking at how Wave 1 PYD measures predicted Wave 2 depression, we also examined how Wave 1 PYD attributes predicted changes in Wave 2 depression scores. In the related analyses, the socio-demographic variables were entered in Step 1. We then entered Wave 1 depression scores in Step 2. Finally, a Wave 1 PYD measure was included in the model in Step 3 [57]. As argued by Steinberg et al. [60], “despite recent advances in structural equation modeling, it is still generally agreed that the use of multiple regression techniques in which one predicts scores on a dependent variable at time 2 while controlling for scores on that same variable at time 1 is an appropriately a conservative strategy”.

## 3. Results

### 3.1. Attrition Analyses

For the attrition analyses, three observations can be highlighted. First, the numbers of Grade 7 and Grade 8 students who dropped out of the study were not big. Second, analyses showed that the participants who dropped out of the study did not differ from those who participated in both waves in terms of the background socio-demographic variables. The only exception is that, compared with the Grade 8 non-dropouts (84.86%), there was a higher proportion of Grade 8 dropouts living with their fathers (90.39%; *χ*^2^ = 5.84, *p* < 0.05, *φ* = 0.06). Third, Grade 7 dropouts displayed higher scores than did non-dropouts on GPYD, PIT, and TPYD measures with small effect sizes; Grade 8 dropouts also showed higher scores on CBC, GPYD, and TPYD measures with small effect sizes. Because the dropouts in both grades were not many and the observed differences between the dropouts and non-dropouts were not large, we concluded that the attrition problem was not a major issue in this study.

### 3.2. Relationships between PYD and Adolescent Depression Based on Pearson Correlation Analyses

*Concurrent correlation coefficients in Wave 1 and Wave 2.* For the concurrent correlation coefficients between PYD measures and depression scores at Time 1 (see Table 2), the results showed that different PYD measures were negatively associated with adolescent depression, including CBC (*r*= −0.33, *p* < 0.001), PA (*r* = −0.27, *p* < 0.001), GPYD (*r* = −0.43, *p* < 0.001), PIT (*r* = −0.40, *p* < 0.001), and TPYD scores (*r* = −0.42, *p* < 0.001). As in Wave 1, we found that different PYD measures were negatively associated with adolescent depression in Wave 2, including CBC (*r* = −0.34, *p* < 0.001), PA (*r* = −0.31, *p* < 0.001), GPYD (*r* = −0.46, *p* < 0.001), PIT (*r* = −0.42, *p* < 0.001), and TPYD scores (*r* = −0.45, *p* < 0.001).

*Longitudinal correlation coefficients between PYD measures in Wave 1 and depression in Wave 2.* Consistent with the findings of the concurrent correlations in Wave 1 and Wave 2, PYD measures in Wave 1 were significantly and negatively correlated with adolescent depression in Wave 2 for all PYD measures (CBC: *r* = −0.22, *p* < 0.001; PA: *r=* −0.17, *p* < 0.001; GPYD: *r* = −0.29, *p* < 0.001; PIT: *r* = −0.27, *p* < 0.001; TPYD: *r* = −0.28, *p* < 0.001).

Taken as a whole, the concurrent and longitudinal correlation analyses provided initial support for Hypothesis 1 and Hypothesis 2.

### 3.3. Regression Analysis Based on PYD and Depression Measures

For the concurrent multiple regression analyses, the findings presented in Table 3 show that after controlling for the covariates, PYD measures in Wave 1 predicted depression in Wave 1, including CBC (*β* = −0.32, *p* < 0.001), PA (*β* = −0.27, *p* < 0.001), GPYD (*β* = −0.43, *p* < 0.001), PIT (*β* = −0.39, *p* < 0.001), and TPYD (β = −0.42, *p* < 0.001). Similar findings were observed for the Wave 2 data, including CBC (*β* = −0.34, *p* < 0.001), PA (*β* = −0.31, *p* < 0.001), GPYD (*β =* −0.47, *p* < 0.001), PIT (*β =* −0.43, *p* < 0.001), and TPYD (*β* = −0.45, *p* < 0.001).

Hierarchical multiple regression analyses were also conducted to examine the influence of Wave 1 PYD measures on Wave 2 depression (see Table 4). After controlling for the Wave 1 covariates, it was found that different PYD measures in Wave 1 significantly predicted Wave 2 depression, including CBC (*β* = −0.21, *p* < 0.001), PA (*β* = −0.17, *p* < 0.001), GPYD (*β* = −0.29, *p* < 0.001), PIT (*β* = −0.26, *p* < 0.001), and TPYD (*β* = −0.28, *p* < 0.001). Finally, we also conducted hierarchical multiple regression analyses to examine the effect of Wave 1 PYD measures on changes in Wave 2 depression scores (Table 4). Besides controlling for the Wave 1 control variables, we also controlled for Wave 1 depression scores to reveal the change in the depression scores in Wave 2. The findings presented in Table 4 show that Wave 1 PYD measures predicted changes in Wave 2 depression scores, including CBC (*β* = −0.07, *p* < 0.001), PA (*β* = −0.06, *p* < 0.01), GPYD (*β* = −0.11, *p* < 0.001), PIT (*β* = −0.10, *p* < 0.001), and TPYD (*β* = −0.11, *p* < 0.001), with small effect sizes.

In summary, concurrent multiple regression analyses provided support for Hypothesis 1 that PYD attributes were linked to adolescent depression at the same time. Longitudinal multiple regression analyses showed that PYD in Wave 1 predicted Wave 2 depression across time, giving support for Hypothesis 2. The prospective analyses controlling for the Wave 1 depression scores also gave support for Hypothesis 3 that PYD attributes in Wave 1 were related to a reduction in depression scores in Wave 2.

## 4. Discussion

With reference to the research gaps in the literature on the relationships between PYD attributes and adolescent depression, there are several unique characteristics of this study. First, the present study was conducted in mainland China with Chinese high school students as participants. Research findings collected from a non-Western context deepen our understanding of the role that culture plays in shaping developmental assets and impacting youth development [26]. This study actively responded to the comment of Wiium and Dimitrova [47] that “little PYD work has been devoted to culturally diverse populations of youth across countries”. Second, as cross-sectional studies are common in this field, a short-term longitudinal study was conducted to look at the longitudinal relationship between PYD and depression across time. Third, the Chinese Positive Youth Development Scale, which generates multiple measures based on 15 primary PYD factors, four higher order factors, and the total score, was employed to look at the influence of PYD from a holistic perspective. Finally, besides the longitudinal association between Wave 1 PYD and Wave 2 depression, we also examined the influence of Wave 1 PYD attributes on changes in adolescent depression over time.

Consistent with our hypotheses, PYD attributes predicted depression scores concurrently and longitudinally. PYD attributes also predicted changes in depression scores over time. Four observations can be highlighted from the present findings. First, these findings are generally in line with the existing literature that PYD attributes, such as resilience, emotional intelligence, self-efficacy, and character strengths, are negatively associated with adolescent depression [61]. Second, the findings are also consistent with the limited literature on the linkages between PYD attributes and depression in Chinese adolescents [62]. Third, positive identity appears to be a significant protective factor against depression in adolescents, which provides insights into the design of PYD programs for Chinese adolescents. The existing youth enhancement programs often over-emphasize cognitive competence for its vital role in achieving academic success but give insufficient attention to establishing positive identity in Chinese adolescents. The present study demonstrates the role of positive identity in promoting adolescent health, which can guide future program design. Fourth, our findings confirmed the important role of family structure in influencing adolescent depression. The results indicated students living in non-intact families were more vulnerable to depressive symptoms than their counterparts during the first year of high school, which was in line with previous research revealing poorer family functioning in non-intact families [63]. In addition, living with the mother was found to be a significant protective factor against depression longitudinally. Our results are in line with previous research highlighting the center role of mothers in parenting in the Chinese context [17,64]. However, as the effect size of the influence was small, there is a need to replicate the findings.

Based on the GPYD and mean total PYD scores, the present findings give general support for the PYD models that show that developmental assets can protect adolescents from depression [26]. Besides, the findings suggest that different aspects of PYD are protective factors for adolescent depression. For example, cognitive–behavioral competence’s negative prediction of depression is consistent with the social–emotional learning framework [30]. Besides, prosocial attributes’ negative prediction of depression is in line with the thesis that external developmental assets (Benson’s view) [26] and contribution (Lerner’s view) [29] are important for adolescent development. Furthermore, the negative prediction of adolescent depression by positive identity measures is in line with the literature on self-esteem and self-efficacy. Theoretically, as a wide range of PYD attributes were employed in this study, the present findings suggest that specific and global PYD attributes are protective factors against adolescent depression.

The present findings also clarify the relationship between eudaimonic well-being and psychological ill-being. In the mental health literature, there are different conceptions of mental health. Traditionally, mental health has been defined in terms of psychological ill-being where symptoms, dysfunction, and personal distress are focused upon. For example, depression is regarded as mental morbidity because there are symptoms that are excessive (such as sadness and sleep) or inadequate (such as loss of motivation and lack of life meaning). Such a conception of mental health has dominated mainstream psychiatry for a long time, as exemplified in the Diagnostic and Statistical Manual of Mental Disorders (DSM-5).

With the emergence of humanistic psychology, existential psychology, and positive psychology, the concept of mental health has shifted from a focus on deficits to the bright side of human beings. For example, Ryff and Keyes [65] suggested that there are several elements of mental health, including autonomy, mastery of the environment, personal development, positive interpersonal relationships, life meaning, and acceptance of oneself. Such a concept is commonly regarded as the eudaimonic concept of well-being [66] with the beliefs that human beings strive for the fulfillment of one’s potential and human development is intrinsically growth-oriented. There are some recent models on the relationship between eudaimonic well-being and health: Boehm and Kubzansky [67] proposed a model on the influence of eudaimonic well-being on cardiovascular disease; Vázquez et al. [68] argued that eudaimonic well-being affects stress and psychosocial resources, which in turn influence the cardiovascular and immune systems and diseases. Comparatively speaking, fewer models link eudaimonic well-being defined by PYD attributes and psychological ill-being indexed by adolescent depression. Based on the existing work on the linkage between eudaimonic well-being and health, it is hypothesized that eudaimonic well-being would influence some biological processes related to stress and psychosocial resources [69], which eventually contribute to psychological morbidity (i.e., negative mental health). Further work on theory development in this area would be helpful.

Practically speaking, the present findings reinforce the view that PYD is a promising approach to protect young people from adolescent depression. In particular, to improve PYD attributes in Chinese adolescents, researchers and practitioners should endeavor to develop and implement culturally adapted curriculum-based PYD programs. For example, a large-scale PYD program in Hong Kong entitled the “Positive Adolescent Training through Holistic Social” (P.A.T.H.S.) program has been implemented in junior high schools in Hong Kong. This project was designed based on solid PYD theories and has been systematically evaluated via different assessment tools [50,70]. In Catalano’s review of worldwide PYD programs, the “P.A.T.H.S. Project” has been listed as one of the effective intervention programs [71]. The Hong Kong “P.A.T.H.S. Project” has been further modified and transplanted to mainland China as the “Tin Ka Ping P.A.T.H.S. Project”. Similarly, Zhu and Shek’s research revealed the positive impacts of the “Tin Ka Ping P.A.T.H.S. Project” on developmental outcomes in students in mainland China [62]. To achieve holistic development in adolescents in China, launching effective and culturally adapted PYD programs will promote PYD attributes and prevent psychological morbidity in adolescents.

This study has several limitations. First, as this is a short-term longitudinal study, it would be helpful to collect more data over a longer period of time to clarify the long-term impact of PYD on adolescent depression. Second, as only four schools were involved, there is a need to collect data from more schools to enhance the generalizability of the findings. Third, as only self-report measures were used in this study, it would be helpful if data based on the significant others of the adolescent participants could be collected. Fourth, among the primary PYD attributes subscales, self-efficacy presented less satisfactory reliability. Although some references suggest that a Cronbach’s alpha value of 0.54 can be acceptable for scales with limited items [53,54], and the values of the mean inter-item correlation and item–total correlation were acceptable, the results should be interpreted with caution. Finally, although similar studies looking at PYD attributes and adolescent depression have been conducted previously, it would be helpful if the mechanisms involved in the impact of PYD on adolescent depression could be clarified in future studies, such as how PYD attributes affect stress processes and psychosocial resources, which would eventually contribute to adolescent depression. Besides, it would be interesting to ask how PYD attributes might affect psychological well-being, which would eventually affect hedonic well-being [72].

## 5. Conclusions

The present study contributes to the field by examining the relationships between PYD attributes and adolescent depression using two-wave longitudinal data in the Chinese context. Four conclusions can be drawn from the findings. First, PYD attributes are negatively related to adolescent depression, both concurrently and longitudinally. Second, PYD attributes predict adolescent depression, both concurrently and longitudinally. Third, higher PYD attributes predict a drop in adolescent depression over time. The present findings underscore the protective effect of PYD attributes against depression in Chinese adolescents.

## Figures and Tables

**Table 1 ijerph-17-04457-t001:** Reliability of measures.

Measures	Wave 1			Wave 2		
	α	Mean Inter-Item Correlation	Mean Item–Total Correlation	α	Mean Inter-Item Correlation	Mean Item–Total Correlation
**PYD primary factors**						
Bonding	0.82	0.43	0.58	0.89	0.57	0.71
Resilience	0.83	0.45	0.59	0.90	0.61	0.73
Social competence	0.84	0.44	0.60	0.89	0.55	0.69
Recognition for positive behavior	0.78	0.48	0.59	0.84	0.58	0.68
Emotional competence	0.83	0.45	0.59	0.87	0.53	0.68
Cognitive competence	0.85	0.48	0.63	0.89	0.59	0.72
Behavioral competence	0.75	0.38	0.51	0.84	0.51	0.64
Moral competence	0.76	0.35	0.50	0.80	0.41	0.56
Self-determination	0.78	0.42	0.55	0.84	0.52	0.65
Self-efficacy	0.54	0.37	0.37	0.59	0.42	0.42
Clear and positive identity	0.84	0.43	0.59	0.87	0.50	0.65
Beliefs in the future	0.75	0.51	0.60	0.79	0.56	0.63
Prosocial involvement	0.82	0.49	0.62	0.88	0.59	0.71
Prosocial norms	0.75	0.38	0.51	0.81	0.47	0.60
Spirituality	0.85	0.46	0.61	0.89	0.53	0.68
**PYD higher-order factors**						
Cognitive–behavioral competence	0.91	0.38	0.58	0.94	0.49	0.68
Prosocial attributes	0.86	0.40	0.57	0.90	0.48	0.66
General PYD qualities	0.95	0.30	0.53	0.96	0.37	0.60
Positive identity	0.88	0.44	0.61	0.90	0.49	0.66
Total PYD	0.97	0.31	0.54	0.98	0.38	0.61
Depression	0.87	0.28	0.49	0.89	0.33	0.54

**Table 2 ijerph-17-04457-t002:** Descriptive and correlational analyses.

Measures	Mean	SD	Correlations															
			1	2	3	4	5	6	7	8	9	10	11	12	13	14	15	16
1. Age	13.12	0.81	--															
2. Gender a			−0.08 ***	--														
3. Family intactness b			0.02	0.01	--													
4. Living with father c			−0.01	0.01	0.29 ***	--												
5. Living with mother c			0.02	−0.02	0.36 ***	0.41 **	--											
6. W1 DP	1.80	0.49	0.08 ***	0.03	0.07 ***	0.07 **	0.06 **	--										
7. W2 DP	1.79	0.52	0.06 **	0.04	0.05 **	0.06 **	0.09 ***	0.47 ***	--									
8. W1 CBC	4.82	0.74	−0.09 ***	−0.01	−0.05 **	−0.05 *	−0.06 **	−0.33 ***	−0.22 ***	--								
9 W1 PA	4.87	0.84	−0.09 ***	0.08 ***	−0.05 *	−0.04	−0.03	−0.27 ***	−0.17 ***	0.66 ***	--							
10. W1 GPYD	4.81	0.71	−0.12 ***	0.01	−0.07 **	−0.05 *	−0.07 ***	−0.43 ***	−0.29 ***	0.83 ***	0.71 ***	--						
11. W1 PIT	4.53	0.93	−0.11 ***	−0.10 ***	−0.07 **	−0.04 *	−0.08 ***	−0.40 ***	−0.27 ***	0.71 ***	0.59 ***	0.74 ***	--					
12. W1 TPYD	4.78	0.69	−0.12 ***	−0.01	−0.07 **	−0.05 *	−0.07 ***	−0.42 ***	−0.28 ***	0.90 ***	0.79 ***	0.97 ***	0.83 ***	--				
13. W2 CBC	4.95	0.78	−0.10 ***	−0.02	−0.02	−0.04 *	−0.05 *	−0.25 ***	−0.34 ***	0.44 ***	0.35 ***	0.43 ***	0.39 ***	0.46 ***	--			
14. W2 PA	5.01	0.85	−0.09 ***	0.07 ***	−0.02	−0.02	−0.05 *	−0.24 ***	−0.31 ***	0.34 ***	0.42 ***	0.40 ***	0.32 ***	0.42 ***	0.66 ***	--		
15. W2 GPYD	4.92	0.76	−0.08 ***	−0.01	−0.03	−0.05 **	−0.08 **	−0.34 ***	−0.46 ***	0.43 ***	0.38 ***	0.49 ***	0.42 ***	0.50 ***	0.84 ***	0.69 ***	--	
16. W2 PIT	4.64	0.95	−0.06 **	−0.12 **	−0.04 *	−0.04 *	−0.04 *	−0.32 ***	−0.42 ***	0.38 ***	0.30 ***	0.41 ***	0.48 ***	0.44 ***	0.72 ***	0.61 ***	0.75 ***	--
17. W2 TPYD	4.90	0.73	−0.09 ***	−0.02	−0.03	−0.05 *	−0.07 **	−0.34 ***	−0.45 ***	0.45 ***	0.41 ***	0.50 ***	0.45 ***	0.51 ***	0.91 ***	0.78 ***	0.97 ***	0.84 ***

Note. ^a^ 1 = male, 2 = female; ^b^ 1 = intact, 2 = non-intact; ^c^ 1 = yes, 2 = no. W1 = Wave 1; W2 = Wave 2; DP = depression; CBC = cognitive–behavioral control; PA = prosocial attributes; GPYD = general positive youth development; PIT = positive identity; TPYD = average of total positive youth development. * *p* < 0.05; ** *p* < 0.01; *** *p* < 0.001.

**Table 3 ijerph-17-04457-t003:** Cross-sectional regression analyses for depression.

Model	Predictors	Depression (Wave 1)					Depression (Wave 2)				
		*β*	*t*	Cohen’s *f2*	R2 Change	F Change	*β*	*t*	Cohen’s *f2*	R2 Change	F Change
1	Age	0.08	3.94 ***	0.006	0.014	7.46 ***	0.06	2.97 **	0.003	0.013	6.75 ***
	Gender a	0.04	1.95	0.001			0.04	2.06 *	0.002		
	Family intactness b	0.05	2.40 *	0.002			0.01	0.60	0.000		
	Living with father c	0.04	1.99	0.002			0.02	1.09	0.000		
	Living with mother c	0.02	0.67	0.000			0.07	3.23 **	0.004		
2	CBC	−0.32	−17.08 ***	0.114	0.099	291.72 ***	−0.34	−17.74 ***	0.126	0.110	314.64 ***
	PA	−0.27	−13.99 ***	0.077	0.073	195.72 ***	−0.31	−16.57 ***	0.109	0.098	274.40 ***
	GPYD	−0.43	−23.74 ***	0.220	0.185	563.76 ***	−0.47	−26.44 ***	0.279	0.215	699.26 ***
	PIT	−0.39	−21.39 ***	0.178	0.153	457.31 ***	−0.43	−23.45 ***	0.220	0.178	550.05 ***
	TPYD	−0.42	−22.99 ***	0.207	0.175	528.58 ***	−0.45	−25.37 ***	0.257	0.201	643.61 ***

Note. In model 2, control variables were statistically controlled for, and predictors were included in the model separately; measures of positive youth development in Wave 1 and Wave 2 were included as predictors to predict depression in Wave 1 and Wave 2, respectively. ^a^ 1 = male, 2 = female; ^b^ 1 = intact, 2 = non-intact; ^c^ 1 = yes, 2 = no. CBC = cognitive–behavioral control; PA = prosocial attributes; GPYD = general positive youth development; PIT = positive identity; TPYD = average of total positive youth development. * *p* < 0.05; ** *p* < 0.01; *** *p* < 0.001.

**Table 4 ijerph-17-04457-t004:** Longitudinal regression analyses for depression.

Model	Predictors	Depression (Wave 2)					Depression (Wave 2)					
		*β*	*t*	Cohen’s *f2*	R2 Change	F Change	*β*	*t*	Cohen’s *f2*	R2 Change	F Change	
1	Age	0.06	2.94 **	0.059	0.014	6.86 ***	0.02	1.20	0.001	0.229	122.82 ***	
	Gender a	0.04	2.08 *	0.041			0.03	1.39	0.001			
	Family intactness b	0.01	0.61	0.012			−0.01	−0.28	0.000			
	Living with father c	0.03	1.15	0.023			0.01	0.25	0.000			
	Living with mother c	0.07	3.25 **	0.065			0.07	3.27 **	0.004			
	W1 Depression						0.47	26.32 ***	0.279			
2	CBC	−0.21	−10.63 ***	0.045	0.043	112.94 ***	−0.07	−3.66 ***	0.005	0.004	13.37 ***	
	PA	−0.17	−8.62 ***	0.004	0.028	74.26 ***	−0.06	−3.04 **	0.004	0.003	9.23 **	
	GPYD	−0.29	−14.98 ***	0.089	0.081	224.24 ***	−0.11	−5.80 ***	0.013	0.010	33.60 ***	
	PIT	−0.26	−13.43 ***	0.072	0.066	180.31 ***	−0.10	−5.00 ***	0.010	0.008	24.96 ***	
	TPYD	−0.28	−14.35 ***	0.082	0.075	205.97 ***	−0.11	−5.42 ***	0.012	0.009	29.38 ***	

Note. In model 2, control variables were statistically controlled for, and predictors measured in Wave 1 were included in the model separately. ^a^ 1 = male, 2 = female; ^b^ 1 = intact, 2 = non-intact; ^c^ 1 = yes, 2 = no. CBC = cognitive–behavioral control; PA = prosocial attributes; GPYD = general positive youth development; PIT = positive identity; TPYD = average of total positive youth development. * *p* < 0.05; ** *p* < 0.01; *** *p* < 0.001.

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
