# Peer review of "Positive Youth Development and Adolescent Depression: A Longitudinal Study Based on Mainland Chinese High School Students"

_ijerph, 2020, doi:10.3390/ijerph17124457_

Round 1

Reviewer 1 Report

I have the following comments for the authors to address:

1) Under the Introduction, the authors stated " Some prominent adolescent mental health issues include depression, suicide and anxiety-related problems." This statement should be improved by providing the evidence e.g. prevalence of suicidal behavior and self-harm in adolescents and associated factors. Please refer to the following landmark study published by IJERPH:

Lim KS, Wong CH, McIntyre RS, et al. Global Lifetime and 12-Month Prevalence of Suicidal Behavior, Deliberate Self-Harm and Non-Suicidal Self-Injury in Children and Adolescents between 1989 and 2018: A Meta-Analysis. Int J Environ Res Public Health. 2019;16(22):4581. Published 2019 Nov 19. doi:10.3390/ijerph16224581

2) The authors stated "The second perspective is the 5C/6C model which highlights the importance of connection, competence, confidence, character, contribution and compassion [28]"Why is 5C or 6c? Should it be 6C?

3) The introduction is too long. Can the authors shorten and make the introduction more concise? For example, the authors can briefly describe 1-13 PYD or summarize it in a table.

4) Under the results, the authors should mention that which PVD primary factor is the most common or highest score.

5) Under the discussion, the authors should mention how to improve PVD of Chinese adolescents. Please specify interventions.

Author Response

I have the following comments for the authors to address:

1) Under the Introduction, the authors stated " Some prominent adolescent mental health issues include depression, suicide and anxiety-related problems." This statement should be improved by providing the evidence e.g. prevalence of suicidal behavior and self-harm in adolescents and associated factors. Please refer to the following landmark study published by IJERPH:

Lim KS, Wong CH, McIntyre RS, et al. Global Lifetime and 12-Month Prevalence of Suicidal Behavior, Deliberate Self-Harm and Non-Suicidal Self-Injury in Children and Adolescents between 1989 and 2018: A Meta-Analysis. Int J Environ Res Public Health. 2019;16(22):4581. Published 2019 Nov 19. doi:10.3390/ijerph16224581

Response: We have added more references in the revised manuscript, including the one suggested by the reviewer.

2) The authors stated "The second perspective is the 5C/6C model which highlights the importance of connection, competence, confidence, character, contribution and compassion [28]"Why is 5C or 6c? Should it be 6C?

Response: We have clarified this point. In the original model, there are 5Cs (connection, competence, confidence, character and care). In the updated model, there are 6Cs (connection, competence, confidence, character, care and contribution).

3) The introduction is too long. Can the authors shorten and make the introduction more concise? For example, the authors can briefly describe 1-13 PYD or summarize it in a table.

Response: We have trimmed the content of the Introduction and summarize the PYD attributes in a table.

4) Under the results, the authors should mention that which PVD primary factor is the most common or highest score.

Response: We have added information on the means in the revised manuscript.

5) Under the discussion, the authors should mention how to improve PVD of Chinese adolescents. Please specify interventions.

Response: We have added more information with particular reference to the Project P.A.T.H.S. and Tin Ka Ping P.A.T.H.S. Project.

Reviewer 2 Report

In the current manuscript, Zhou et al attempted to establish a link between positive youth development (PYD) attributes and psychological morbidity, particularly focusing on depression in a relatively large cohort of Chinese adolescents. The authors carried out a longitudinal study in which data from 2,647 junior Chinese high school students were collected in two waves and subsequently analyzed for measures of well-being and depression using various validated psychometric scales. Much of the data revealed that PYD measures at Wave 1 predicted Wave 2 depression scores negatively and that those measures were negatively related to adolescent depression over time. The authors concluded by saying that PYD attributes were pivotal in protecting adolescents from depression.

Reading through the manuscript was easy, and the presentation of results were straightforward and apt. Section 1 (Introduction) established a nice flow, right from introducing the problem to the revelation of the scientific questions that were addressed in the study. However, the introduction felt a tad too long, to an extent that it was difficult to keep track of all the information which at times instilled boredom. Section 2 (Materials and Methods) was described well including the description of the statistical analysis plan. Section 3 (Results) addressed the questions asked and is satisfactory. Sections 4 and 5 (Discussion and conclusion) discussed the results well in the context of the current findings and already published literature, yet, it felt like the authors were trying to be too assertive on issues like ‘sample-size’, for example. The fact that the sample size in their current work was larger than earlier published work seemed to be emphasized at several sections of the manuscript. Moreover, the authors should try and refrain themselves from using statements like ‘……pioneering study in the field of PYD…..’ (line 483) since it undermines earlier published work on PYD attributes.

Overall, I think the manuscript is well written and is suitable for publication in the current journal after revision. My suggestion would be to work on the introduction and try to make it concise, so that a naïve reader can easily understand the problems in the field and why was this study important to address those issues.

I have some scientific questions on the current manuscript.

Major questions

  1. In lines 59-63, the authors cite a paper that pointed out that children in Asian countries were more likely to suffer from depression than those in the Western countries. This information intrigues me. First, is there any work done that addresses differences in sub-threshold depression between Asia vs. the West? If there is, it would be good to cite those in the current manuscript. Second, is there evidence explaining why Asian children are prone to have depressive symptoms in comparison to their western counterparts? Apart from the socioeconomic differences, some of which are discussed in the manuscript, it would be very interesting to see if there are genetic attributes that account for these differences? Any discussion on genotype vs. phenotype, or genotype vs. environment in generating differences between depression in Asia vs. the West could make the introduction more engaging to read.
  2. In lines 263-264, it is stated that there were 401 students who reported that their families were not intact, or that their parents were separated. Were these students included in the study? This information is not clear in the manuscript. If they were included, the authors should discuss those results and potentially introduce published work that talk about the effects of parental separation on mental health in adolescents.
  3. In lines 290-291, the alpha value for the self-efficacy subscale was reported to be 0.54. Although the authors considered this to be acceptable (since there were two items), I think it still points towards a more erroneous value. The authors should be careful in describing and interpreting this data and come up with an explanation on the inclusion of this subscale in the dataset.

Minor questions

  1. In line 44, the authors cite Thapar et al, which reported that adolescent depression exceeded 4% in underdeveloped and developing countries. What do the authors mean by this? Please, clarify.
  2. Although the authors clearly mention towards the end of their introduction that cross-sectional studies on PYD attributes and mental health are aplenty, yet their first hypothesis hovers around cross-sectional associations. For me, this analysis seems redundant.

General comments

  1. Please maintain a standard/consistent way of citing references. There are several discrepancies in the way citations are managed in the whole manuscript. For example, in line 297, Wang et al., 2013 suddenly pops up although references are numbered in the manuscript. In line 328, page number of a reference is cited which is not necessary. In line 65, please remove e.g. before the reference.
  2. Please shorten the introduction and make it more concise, engaging and pleasing for the reader.
  3. There are minor spelling mistakes and grammatical errors in portions of the manuscript. Please, correct them.

Author Response

In the current manuscript, Zhou et al attempted to establish a link between positive youth development (PYD) attributes and psychological morbidity, particularly focusing on depression in a relatively large cohort of Chinese adolescents. The authors carried out a longitudinal study in which data from 2,647 junior Chinese high school students were collected in two waves and subsequently analyzed for measures of well-being and depression using various validated psychometric scales. Much of the data revealed that PYD measures at Wave 1 predicted Wave 2 depression scores negatively and that those measures were negatively related to adolescent depression over time. The authors concluded by saying that PYD attributes were pivotal in protecting adolescents from depression.

Reading through the manuscript was easy, and the presentation of results were straightforward and apt. Section 1 (Introduction) established a nice flow, right from introducing the problem to the revelation of the scientific questions that were addressed in the study. However, the introduction felt a tad too long, to an extent that it was difficult to keep track of all the information which at times instilled boredom. Section 2 (Materials and Methods) was described well including the description of the statistical analysis plan. Section 3 (Results) addressed the questions asked and is satisfactory. Sections 4 and 5 (Discussion and conclusion) discussed the results well in the context of the current findings and already published literature, yet, it felt like the authors were trying to be too assertive on issues like ‘sample-size’, for example. The fact that the sample size in their current work was larger than earlier published work seemed to be emphasized at several sections of the manuscript. Moreover, the authors should try and refrain themselves from using statements like ‘……pioneering study in the field of PYD…..’ (line 483) since it undermines earlier published work on PYD attributes.

Response: We have trimmed the Introduction. We have also toned down the assertions in the Discussion section.

Overall, I think the manuscript is well written and is suitable for publication in the current journal after revision. My suggestion would be to work on the introduction and try to make it concise, so that a naïve reader can easily understand the problems in the field and why was this study important to address those issues.

Response: We have trimmed the Introduction.

I have some scientific questions on the current manuscript.

Major questions

1. In lines 59-63, the authors cite a paper that pointed out that children in Asian countries were more likely to suffer from depression than those in the Western countries. This information intrigues me. First, is there any work done that addresses differences in sub-threshold depression between Asia vs. the West? If there is, it would be good to cite those in the current manuscript. Second, is there evidence explaining why Asian children are prone to have depressive symptoms in comparison to their western counterparts? Apart from the socioeconomic differences, some of which are discussed in the manuscript, it would be very interesting to see if there are genetic attributes that account for these differences? Any discussion on genotype vs. phenotype, or genotype vs. environment in generating differences between depression in Asia vs. the West could make the introduction more engaging to read.

Response: We have added information on this point. Some possible cultural factors leading to the observed phenomenon are briefly discussed. However, it is noteworthy that there are not many comparative studies in this area.

2. In lines 263-264, it is stated that there were 401 students who reported that their families were not intact, or that their parents were separated. Were these students included in the study? This information is not clear in the manuscript. If they were included, the authors should discuss those results and potentially introduce published work that talk about the effects of parental separation on mental health in adolescents.

Response: These students were included in the study. Actually, family intactness was treated as a covariate in the study.

3. In lines 290-291, the alpha value for the self-efficacy subscale was reported to be 0.54. Although the authors considered this to be acceptable (since there were two items), I think it still points towards a more erroneous value. The authors should be careful in describing and interpreting this data and come up with an explanation on the inclusion of this subscale in the dataset.

Response: We have clarified this point further. trimmed the Introduction. We have also added this limitation in the Discussion section.

Minor questions

1. In line 44, the authors cite Thapar et al, which reported that adolescent depression exceeded 4% in underdeveloped and developing countries. What do the authors mean by this? Please, clarify.

Response: We have clarified this in the revised manuscript.

2. Although the authors clearly mention towards the end of their introduction that cross-sectional studies on PYD attributes and mental health are aplenty, yet their first hypothesis hovers around cross-sectional associations. For me, this analysis seems redundant.

Response: While cross-sectional studies are plenty in the West, they are very few in Chinese contexts. We have clarified this in the revised manuscript.

General comments

1. Please maintain a standard/consistent way of citing references. There are several discrepancies in the way citations are managed in the whole manuscript. For example, in line 297, Wang et al., 2013 suddenly pops up although references are numbered in the manuscript. In line 328, page number of a reference is cited which is not necessary. In line 65, please remove e.g. before the reference.

Response: We have rectified this and checked the revised manuscript carefully.

2. Please shorten the introduction and make it more concise, engaging and pleasing for the reader.

Response: We have trimmed the Introduction.

3. There are minor spelling mistakes and grammatical errors in portions of the manuscript. Please, correct them.

Response: We have rectified the related errors and checked the English carefully.